# Development and Internal Validation of Risk Assessment Models for Chronic Obstructive Pulmonary Disease in Coal Workers

**DOI:** 10.3390/ijerph20043655

**Published:** 2023-02-18

**Authors:** Hui Wang, Rui Meng, Xuelin Wang, Zhikang Si, Zekun Zhao, Haipeng Lu, Huan Wang, Jiaqi Hu, Yizhan Zheng, Jiaqi Chen, Ziwei Zheng, Yuanyu Chen, Yongzhong Yang, Xiaoming Li, Ling Xue, Jian Sun, Jianhui Wu

**Affiliations:** School of Public Health, North China University of Science and Technology, No. 21 Bohai Avenue, Caofeidian New Town, Tangshan 063210, China

**Keywords:** coal workers, risk assessment model, risk scoring system, chronic obstructive pulmonary disease, random forest model

## Abstract

Coal workers are more likely to develop chronic obstructive pulmonary disease due to exposure to occupational hazards such as dust. In this study, a risk scoring system is constructed according to the optimal model to provide feasible suggestions for the prevention of chronic obstructive pulmonary disease in coal workers. Using 3955 coal workers who participated in occupational health check-ups at Gequan mine and Dongpang mine of Hebei Jizhong Energy from July 2018 to August 2018 as the study subjects, random forest, logistic regression, and convolutional neural network models are established, and model performance is evaluated to select the optimal model, and finally a risk scoring system is constructed according to the optimal model to achieve model visualization. The training set results show that the logistic, random forest, and CNN models have sensitivities of 78.55%, 86.89%, and 77.18%; specificities of 85.23%, 92.32%, and 87.61%; accuracies of 81.21%, 85.40%, and 83.02%; Brier scores of 0.14, 0.10, and 0.14; and AUCs of 0.76, 0.88, and 0.78, respectively, and similar results are obtained for the test set and validation set, with the random forest model outperforming the other two models. The risk scoring system constructed according to the importance ranking of random forest predictor variables has an AUC of 0.842; the evaluation results of the risk scoring system shows that its accuracy rate is 83.7% and the AUC is 0.827, and the established risk scoring system has good discriminatory ability. The random forest model outperforms the CNN and logistic regression models. The chronic obstructive pulmonary disease risk scoring system constructed based on the random forest model has good discriminatory power.

## 1. Introduction

Chronic obstructive pulmonary disease (COPD) is a common preventable respiratory disease characterized by persistent airflow limitation, which is associated with an increased chronic inflammatory response of the airways and lungs to toxic particles or gases. COPD has a high prevalence and mortality, and it is the third leading cause of death worldwide; the global prevalence of COPD in 2019 was 13.1%, with prevalence rates ranging from 11.6% to 13.9% in different regions of the world [1]. COPD not only affects lung function but also has extrapulmonary effects that affect the whole body, with common comorbidities including cardiovascular disease, lung cancer, osteoporosis, anxiety, and depression [2]. COPD has serious health hazards for individuals and there is no effective way to slow down the progression of the disease in the present. Once the condition of COPD patients deteriorates, not only will their lung function level decrease, but also increase the mortality rate and disability rate [3]. Smoking, air pollution, biomass fuels, and occupational dust exposure are considered to be important risk factors for COPD. Due to the particularity of the working environment, coal workers are often exposed to dust, chemical substances, and other occupational harmful factors, which increase the risk of COPD [4]. At present, the research on COPD is mainly based on the general population to understand the pathogenesis and influencing factors of COPD, and there are few studies on coal workers.

Risk assessment models based on machine learning algorithms for related diseases have been widely used in the medical field [5,6]. Commonly used machine learning algorithms mainly include logistic regression, random forest, Xgboost, and convolutional neural network, and each algorithm has its own advantages and disadvantages. As an ensemble algorithm composed of multiple decision trees, random forest can be better applied to large datasets and has better prediction performance than a single estimator. The logistic regression model is a simple and highly interpretable model, but it cannot handle the complex relationship between the independent variables and the dependent variables, and it is easy to underfit and the accuracy is not high. Compared with general neural networks, convolutional neural networks can effectively reduce the complexity of the model by using weight sharing and sparse connection, and CNN (convolutional neural networks) is widely used in medical image recognition [7,8].

At present, risk assessment models for COPD mainly assess the risk of hospitalization of COPD patients due to deterioration of the condition [9], and there are few models that assess the risk of COPD in occupational populations. Therefore, in order to protect the lung health of coal workers, we urgently need to establish a COPD risk assessment model suitable for coal workers, and establish a risk scoring system according to the optimal model.

## 2. Materials and Methods

### 2.1. Research Object

This study relies on China’s key Research and Development program “Cohort Study on Health Effects of Occupational Groups in Beijing-Tianjin-Hebei Region”, and 3955 coal workers who participated in occupational health examinations in Gequan Mine and Dongpang Mine in Hebei province from July 2018 to August 2018 are the research objects.

Inclusion criteria: 18~60 years old, ≥1 year of service. Exclusion criteria: those who could not measure lung function, i.e.,: those who had undergone chest, abdominal, or eye surgery in the past 3 months, those who were pregnant or breastfeeding, and those who had been hospitalized for heart disease in the past 1 month; those who had missing information from the questionnaire.

The study was conducted in accordance with the Declaration of Helsinki, verified and approved by the Ethics Committee of the North China University of Technology (15006), and all study subjects voluntarily participated in this investigation and signed an informed consent form.

### 2.2. Data Collection

Personal information was obtained through questionnaires, which are administered to workers by professional staff in a one-to-one manner. The content of the questionnaire mainly includes the following sections: (1) demographic information: age, gender, ethnicity, marital status, education level, economic income, etc.; (2) behavioral lifestyle: smoking, drinking status, dietary conditions, physical activity, sleep quality; (3) personal history of diseases: hypertension, diabetes, tumors; (4) work status: nature of employment, length of service, type of work, shift situation.

### 2.3. Physical Examination

(1)Height and weight: measurements were obtained with the Dekang DK-08-C height and weight meter, for which the subjects should remove shoes, hats, watches, and other items that affect the test results and the measurements should be obtained in the correct position according to the instructions of the relevant personnel.(2)Pulmonary function test: pulmonary function measurements were obtained as instructed by the staff, where the subject should sit quietly, sit with the upper body straight, keep the head horizontal, clip on the nose clip, and put on the mouthpiece according to the instructions of the professional staff before the test, while ensuring that the tongue cannot block the mouthpiece or leak air.

### 2.4. Definition of Ending

The pulmonary function test was performed by professionals using a portable spirometer (China CHEST) to measure mainly the first and second expiratory volume with force (FEV_1_), force spirometry (FVC), and according to the 2017 Global Initiative for Chronic Obstructive Lung Disease (GOLD) guidelines [9], FEV_1_/FVC < 70% is diagnosed as COPD.

### 2.5. Variable Definitions

#### 2.5.1. Body Mass Index(BMI)

BMI = weight (kg)/height^2^(m^2^), BMI < 18.5 kg/m^2^ is defined as underweight, 18.5 kg/m^2^ ≤ BMI < 24 kg/m^2^ is defined as normal weight, 24 kg/m^2^ ≤ BMI < 28 kg/m^2^ is defined as overweight, and BMI ≥ 28 kg/m^2^ is defined as obese.

#### 2.5.2. Smoking Index (SI)

Smoking index = daily smoking index × number of years of smoking, grouped as 0, 1, 100~, and 200~.

#### 2.5.3. Drinking Status

In this study, drinking status was categorized as never drinking, formerly abstained from drinking, and current drinking.

#### 2.5.4. Physical Exercise

Exercise was determined by exercising more than 3 times a week and for more than half an hour each time.

#### 2.5.5. Physical Activity

In this study, the International Physical Activity Questionnaire (IPAQ) was used to investigate the physical activity of coal workers [10]. Physical activity was classified as “low”, “medium”, and “high” according to intensity, frequency, and overall weekly physical activity level. The overall weekly physical activity level < 600 MET-min/w is considered low, the overall weekly physical activity level 600 to MET-min/w is considered medium, and the overall weekly physical activity level 3000~ MET-min/w is considered high.

#### 2.5.6. Sleep Quality

The Athens Insomnia Scale (AIS) was applied to assess the sleep quality of coal workers [11], with scores <4 being accessibility, with scores 4–6 being suspected insomnia and scores >6 being insomnia.

#### 2.5.7. Cumulative Dust Exposure (CDE)

The criteria for determining dust exposure in this study are based on the “Determination of Dust in Workplace Air Part 1: Total Dust Concentration”, and the cumulative individual dust exposure is calculated based on the total dust concentration in the workplace measured by a qualified testing company and the actual results of daily testing [12].
(1)$$CDE=C_{1}*T_{1}+C_{2}*T_{2}+C_{3}*T_{3}+...+C_{n}*T_{n}$$

C_n_ is the annual geometric mean concentration in mg/m^3^ for a job performed by a coal worker; T_n_ is the duration of dust pick-up in years for a job performed by a worker. The specific grouping is as follows: <50, 50~, and 100~.

#### 2.5.8. Shift Situation

A system of working hours in which the production process requires 24 h of continuous work, guaranteed by one or several teams working in shifts determines the shift situation. This study classifies shift work situations into the following three situations, never shifted, ever shifted, and now shifted [13].

#### 2.5.9. Ventilation and Dust Removal Measures

The evaluation of ventilation and dust removal measures were combined with the evaluation results of the inspection company and the evaluation of the operation of the facility in the daily work of coal workers. The specific grouping is as follows: difference, ordinary, and good.

### 2.6. Statistical Methods

The counts were expressed as rates, and the chi-square test was used for comparison between groups; unconditional logistic regression was used for multi-factor analysis. Through a large number of literature review and collection of relevant data, univariate analysis of relevant factors was carried out, and the variables meaningful for univariate analysis were further incorporated into multivariate analysis, and the influencing factors of COPD of coal workers were finally determined. The statistical tests were all two-sided, and the test level was α = 0.05. All of this was carried out in the SPSS 22.0 statistical software (IBM, Armonk, NY, USA).

### 2.7. Model Establishment

In this study, sklearn.model_selection.train_test_split was used to divide the dataset into training set, test set, and validation set according to 7:2:1 (Appendix A). The screening of model predictors was carried out through univariate analysis, multivariate analysis, and literature review to construct a risk assessment model.

Logistic regression model is a classification algorithm that uses a sigmoid function for classification and is implemented in this study using the Sklearn. Logistic Regression module (Appendix A).

The convolutional neural network model consists of convolutional layer, pooling layer, activation layer, and finally a fully connected layer for the classification output. In this study, the CNNs were constructed using keras, the activation function is Relu, the loss function is binary_crossentropy and the optimizer is rmsprop (Appendix A).

Random forest model is essentially a collection of multiple decision trees and is an ensemble learning method. The random forest model is built using the Random Forest Classifier module in sklearn, and the parameters are tuned by the learning curve and the grid search method RandomizdSearchCV. In this model, the following parameters were adjusted, including the tree tree n_estimators estimators, the maximum depth of the tree max_depth, the number of randomly selected features max_festures, and the minimum number of samples min_samples_split, in order to ensure a good learning ability and generalization ability to avoid overfitting (Appendix A).

All models are built in Python 3.10.

### 2.8. Model Evaluation

The performance of the model was evaluated in terms of both discrimination and calibration.

Discrimination is a measure of a model’s ability to distinguish between patients and non-patients and, and commonly evaluated metrics include sensitivity, specificity, accuracy, ROC curve, and its area under the curve AUC.
(2)$$\mathrm{Sensitivity}=\frac{TP}{TP+FN}$$
(3)$$\mathrm{Accuracy}=\frac{TP+TN}{TP+FP+TN+FN}$$
(4)$$\mathrm{Specifity}=\frac{TN}{TN+FP}$$

Calibration is a measure of the accuracy of a model in assessing the future occurrence of an outcome event for an individual, and commonly used measures are Brier score and calibration curve. Calibration curve is an important method to evaluate the calibration of a model, it can visually measure the consistency between the predicted probability and the true probability of the model; the closer the curve is to the diagonal line means the better the calibration of the model.

### 2.9. Establishment of a Risk Scoring System

The optimal model was derived from the development and evaluation of a COPD risk assessment model for coal workers, on the basis of which a risk scoring system was established.

#### 2.9.1. Risk Scoring System

A risk scoring system was constructed using an assignment method based on the importance ranking of the optimal model predictor variables, which involve being assigned in the following manner.

The hazard score corresponding to each independent variable $S_{n}$ is the relative importance of the respective variable $I_{n}$ divided by the smallest relative importance $I_{m}$, i.e.,
(5)$$S_{n}\approx \frac{I_{n}}{I_{m}}$$

Total hazard fraction S_c_ is the sum of the individual hazard scores, i.e.,
(6)$$S_{c}=S_{1}+S_{2}+S_{3}+...+S_{n}$$

Combining the results of the single-factor and multi-factor analyses, the risk factors are set to a maximum value and the protective factors are set to a minimum value of 0. The risk scores for each factor is displayed in the results section of the risk scoring system.

①When the variable is dichotomous, it is assigned to 0, S_n_;②When the variable is a trivial variable, it is assigned to 0, S_n_/2, S_n_;③When the variable is a four-category variable, it is assigned as 0, S_n_/3, 2S_n_/3, S_n_;

#### 2.9.2. Mapping the ROC Curve of a COPD Risk Scoring System for Coal Workers

A risk scoring system was constructed by randomly selecting 70% of the participants, and an ROC curve is drawn according to their score and whether they have COPD.

#### 2.9.3. Setting up Hazard Stratification

According to the ROC curve of the COPD risk scoring system, the maximum M of the Jordon index was found on the ROC curve, and the study subjects were divided into two levels: low-risk population (S_c_ < M) and high-risk population (S_c_ ≥ M).

#### 2.9.4. Performance Evaluation of Risk Scoring Systems

The remaining 30% of workers, classified according to the above classification criteria, were used to calculate the accuracy rate of the risk scoring system.The area under the ROC curve was used to determine the diagnostic value of the risk scoring system.

The area under the ROC curve ≤ 0.5 indicates that the risk scoring system has no diagnostic value. The area under the ROC curve 0.5~0.7 indicates that the risk scoring system has diagnostic value. The area under the ROC curve 0.7~0.8 indicates that the risk scoring system has good diagnostic value. The area under the ROC curve > 0.8 indicates that the diagnostic value of the risk scoring system is sufficient, and the sensitivity and specificity of the risk scoring system are high, which can better identify for disease.

### 2.10. Quality Control

Pre-survey training was provided to investigators and information entry for the questionnaire was carried out in pairs to ensure the accuracy of the data. When performing pulmonary function measurement, staff should instruct participants to perform measurements in accordance with standard movements to ensure the quality of pulmonary function test and increase the accuracy and reliability of outcome diagnosis. Factor analysis and review of the literature ensured that factors associated with outcomes were included in the model and that appropriate statistical analysis methods were used.

## 3. Results

### 3.1. Analysis of General Demographic Characteristics

This study includes 3955 study participants, of which 918 coal workers have COPD, with a prevalence rate of 23.2%. A univariate analysis of the relationship between general demographic characteristics of coal workers and COPD shows that age, gender, education, household income, and BMI are all associated with COPD, with statistically significant differences (*p* < 0.05), as detailed in Table 1

### 3.2. Analysis of the Health Status of Coal Workers

The univariate analysis of the relationship between the health status of coal workers and COPD shows that the personal history of respiratory diseases is associated with COPD, and the difference is statistically significant (*p* < 0.05), as detailed in Table 2

### 3.3. Lifestyle Analysis of Coal Worker Behavior

Through the univariate analysis of the relationship between coal workers’ behavior and lifestyle and COPD, the result shows that smoking index, physical exercise, vegetable intake, and fruit intake are all related to COPD, and the differences are statistically significant (*p* < 0.05), as detailed in Table 3.

### 3.4. Analysis of Occupational Harmful Factors of Coal Workers

A univariate analysis of the relationship between occupational factors and COPD in coal workers showed that seniority, cumulative dust exposure, ventilation, and dust removal measures, mask usage and chemical poison exposure are all associated with COPD, with statistically significant differences (*p* < 0.05); see Table 4 for details.

### 3.5. Multivariate Analysis of Influencing Factors of COPD among Coal Workers

The meaningful influencing factors of univariate analysis were used as input variables to perform unconditional logistic regression analysis for coal workers’ COPD, and the assignment method is shown (Table 5). Multicollinearity diagnosis of independent variables requiring inclusion in multivariate analysis shows (Table 6) that variance inflation factors (VIF) are greater than 0 and less than 10, and a tolerance greater than 0.1 for all variables. The result shows (Table 7) that age 30 and above, male, history of respiratory diseases, smoking index 1 and above, cumulative dust exposure 50 and above, working experience of 10 years and above, and exposure to chemical poisons are risk factors for COPD in coal workers (all *p* < 0.05), and with a bachelor’s degree (junior college) or above, physical exercise, and from3–4 days/week to the daily use of masks along with generally good ventilation and dust removal measures are protective factors for the occurrence of COPD in coal workers (all *p* < 0.05).

### 3.6. Model Results

According to the result of the multi-factor analysis and literature review, a risk assessment model was constructed by including age, gender, education level, personal history of respiratory diseases, smoking index, physical exercise, seniority, mask usage, ventilation and dust removal measures, cumulative dust exposure, and chemical poison exposure.

In the training set (Table 8), the sensitivity, specificity, accuracy, and AUC of random forest are 86.89%, 92.32%, 85.40%, and 0.88, respectively, which are higher than those of the CNN and logistic models. The Brier score and Log loss of random forest are 0.10 and 0.35, respectively, which are lower than those of the CNN and logistic models, and the random forest model has the best performance.

In the test set (Table 8), the sensitivity, specificity, accuracy, and AUC of random forest are 81.86%, 87.06%, 85.10%, and 0.82, respectively, which are higher than those of the CNN and logistic models. The Brier score and Log loss of random forest are 0.13 and 0.41, respectively, which are lower than those of the CNN and logistic models, and the random forest model has the best performance.

In the validation set (Table 8), the sensitivity, specificity, accuracy, and AUC of random forest are 82.93%, 84.30%, 83.11%, and 0.78, respectively, which are higher than those of the CNN and logistic models. The Brier score and Log loss of random forest are 0.11 and 0.37, respectively, which are lower than those of the CNN and logistic models, and the random forest model has the best performance.

The calibration curve of the random forest (Figure 1a–c) is closer to the diagonal line, indicating that the model’s predicted value is closer to the true value. The ROC curve (Figure 2a–c) shows that the random forest model outperforms the other two models in all three sets.

In summary, the random forest model outperforms the CNN and logistic models in the risk assessment of COPD in coal workers.

The optimal model is the random forest model and the variables are ranked in importance according to the optimal model. The result is shown in Figure 3, where chemical poison exposure, cumulative dust exposure, mask usage, and smoking index are the important predictor variables for the random forest model.

### 3.7. Risk Scoring System

Based on the model evaluation, the optimal model is the random forest model, on which the risk scoring system is constructed. The risk scoring system was constructed using the assignment method according to the importance of the predictor variables (Figure 3), and the assignment method is shown in Table 9.

A risk scoring system was constructed for a random sample of 70% of the study subjects and ROC curves were plotted according to their scores and whether they have COPD, the results of which are shown in Figure 4, with an AUC of 0.842. Risk stratification was set: a risk score of 23.05 has the highest Jorden index; therefore, a risk score < 23.05 is defined as low risk and a risk score ≥ 23.05 as high risk.

The remaining 30% of the study subjects was used to evaluate the performance of the risk scoring system. The study population was assigned a risk score according to Table 8 and classified according to the classification criteria. The result shows (Table 10) that 774 people in the low-risk group are normal and 52 have COPD, and 141 of the high-risk population are normal and 220 have COPD. The accuracy of the risk scoring system is 83.7%, and the AUC of the ROC curve is 0.827 (Figure 5), indicating that the established risk scoring system has good discriminating ability.

## 4. Discussion

Coal meets 27% of the world’s energy needs, supplies 40% of the world’s electricity, and is an important pillar of China’s industry [14]. A large number of coal workers are exposed to dust, noise, vibration, and high heat, which can lead to occupational diseases such as pneumoconiosis, noise deafness, vibration sickness, and various chronic diseases [15,16]. Our study is dedicated to the physical health of coal workers and we have constructed a risk assessment model and a risk scoring system suitable for COPD in coal workers.

A total of 3955 coal workers were included in the study, with a COPD prevalence rate of 23.2%, which is higher than that of the general population [17]. Older age was a risk factor for COPD in this study, with an OR of 1.770 (1.063–2.948), which is consistent with the result of related study [18]. This may be related to lung ageing, reduced lung function, and reduced immunity of the lungs to environmental injury [19]. The study found that being male is a risk factor for COPD, with an OR of 3.965 (2.172–7.247). The higher risk of disease in males may be due to the fact that male coal workers are more likely to smoke, but there is also a study that suggests the risk of COPD in females is increasing [20]. This may be related to women’s greater exposure to biomass fuels, higher sensitivity to cigarette smoke, and a faster decline in FEV_1_ in female smokers [21,22]. This study focuses on coal workers, who are far more male than female, so there may be some bias in the investigation of the effect of gender on COPD. Personal history of respiratory disease is a risk factor for COPD in this study, and it mainly refers to a history of tuberculosis and asthma. Asthma is an important cause of the acceleration of FEV_1_ reduction [23]. Tuberculosis is an important cause of airflow obstruction and respiratory symptoms [24,25]. The result of this study, which quantifies smoking in coal workers using a smoking index, suggests that smoking is a risk factor for COPD, which has been considered a major risk factor for COPD in many previous studies [26,27]. This may be due to the fact that cigarette smoke stimulates the release of inflammatory cytokines from respiratory cells, leading to respiratory damage [28,29]. Dust is an important occupational factor for coal workers, and this study quantifies the dust exposure of coal workers by using cumulative dust exposure. The OR values of cumulative dust exposure exceeding 50 mg/m^3^ and 100 mg/m^3^ per year are 1.382 (1.039–1.837) and 2.228 (1.638–3.029), respectively, and the increase in cumulative dust exposure will lead to an increased risk of COPD among coal workers. The possible reason for this is that coal dust can inactivate α-1 antitrypsin and produce reactive oxygen species, that α-1 antitrypsin inactivation increases the risk of COPD, and that reactive oxygen species may lead to emphysema in miners [30]. Seniority refers to the number of years of exposure to dust, and in this study 10 years or more of service can lead to an increased risk of COPD among coal workers. Exposure to chemical poison is also an occupational hazard for coal workers, that mainly refers to inhalation of irritant gases and fumes. Chemical poison exposure usually activates alveolar macrophages and leukocytes, leading to the release of reactive oxygen species, which leads to inflammatory changes in the airways and increases the risk of COPD [31]. Masks and ventilation and dust removal measures are important dust prevention measures for coal workers, and in this study, they are protective factors that can reduce the risk of COPD among workers [32]. These protective measures are important in a high-risk environment such as coal mines to achieve primary prevention of occupational diseases. Physical exercise is a protective factor in this study, and those who carry out physical exercise have a lower risk of COPD, suggesting that increasing physical exercise among coal workers can reduce the decline in FEV_1_ [33]. Previous studies have found that physical activity is the strongest predictor of all-cause mortality in COPD patients [34]. It is also an important measure of pulmonary rehabilitation in COPD patients [35]. The level of education above a bachelor’s degree is a protective factor for COPD, which may be associated with good lifestyle habits and minimal dust exposure in those with high levels of education [36].

In this study, the dataset was divided into three sets: training set, test set, and verification set; and three models of logistic, random forest, and convolutional neural network were established to evaluate the risk of COPD in coal workers. The performance of the models was evaluated from the aspects of discrimination and calibration. The results show that the random forest model has the best performance, with a sensitivity of 81.86% (test set) and a specificity of 87.06% (test set), which is more suitable for the risk assessment of COPD in coal workers. The random forest model is an improvement on the decision tree model that is widely used in the medical field and outperforms other models in some studies [37,38]. In this study, the CNN is better than the logistic model but not as good as the random forest model. In one study, Sandeep Bodduluri uses a machine learning algorithm to distinguish between the structural phenotypes of slow-onset lungs, in which the AUC of CNN and random forest models are 0.80 and 0.78, respectively, and CNN performs better [39]. CNN performs differently in different studies, which may be related to the type of data. CNN achieves better results in the recognition of images, and the application effect in other areas varied depending on the data. The logistic model performs the worst in this study, indicating that the model’s predicted values deviate significantly from the actual values and is not suitable for the risk assessment of COPD in coal workers. The importance ranking of the predictors of the random forest model indicates that chemical poison exposure, cumulative dust exposure, mask usage, smoking index, and ventilation and dust removal measures are important predictors, and the importance ranking of predictors indicates measures that coal workers can employ to achieve higher health benefits. This study constructs a risk scoring system for COPD based on the importance ranking of the optimal model random forest predictor variables and evaluates the risk scoring system with an accuracy of 83.7% and an AUC of 0.827, indicating that the scoring system has good discriminatory ability. The establishment of the risk scoring system explores the application value of the model, which can calculate the individual risk score according to their health data, evaluate the risk of COPD of individual occurrences, and provide a reference basis for the health management of coal workers.

There are some limitations of this study. First, biomass fuels and air pollution are also important influence factors of COPD [40]. However, due to the design of the questionnaire and the collection of the samples, we lack data on this component, so we are unable to include these two variables in the study. In addition, this is a cross-sectional study and therefore inferior to prospective cohort studies in verifying causality. Due to data collection limitations, we did not include coal workers over 60 years of age, which may have led to selective bias. Follow-up studies can survey retired workers to assess the effect of age on coal workers’ COPD. In this study, we did not stage COPD, taking into account the use of the model and the distribution of pulmonary function test data. If COPD is not staged, it may be difficult to extrapolate the model because of differences in the distribution of data. The contribution of our study is mainly to provide a risk assessment model for COPD in coal workers and to construct a risk scoring system based on the risk assessment model. As pulmonary function testing is low among coal workers in daily life, our risk scoring system can be used to assess the risk of COPD among coal workers without pulmonary function testing using health check-up data, and to make targeted recommendations based on the individual’s relevant circumstances, thereby protecting the health of coal workers. The innovation of this paper lies in the fact that, firstly, our research is based on the data obtained from field surveys to explore the relevant influencing factors of the disease, then we used the obtained data to build a risk assessment model suitable for the research object, and finally we realized the model visualization by building a risk scoring system, which increases the applicability of the model.

According to the conclusions of our COPD study of coal workers, the following measures can effectively reduce the occupational hazards of coal dust for coal workers. Ventilation and dust removal measures are important protective measures, so water injection into coal seams, the adoption of new dust prevention technologies, and ensuring the good functioning of ventilation systems in the workplace can help reduce coal workers’ dust exposure. Carrying out health education for coal workers, strengthening workers’ awareness of dust prevention and the use of masks, and encouraging workers to develop healthy lifestyle habits, such as quitting smoking and exercising, are all important measures.

## 5. Conclusions

In this study, the analysis of the relevant data of coal workers shows that an age 30 years old and above, male, personal history of respiratory diseases, smoking index 1 and above, cumulative dust exposure 50 mg/m^3^ and above, seniority ≥ 10 years, and exposure to chemical poison are risk factors for COPD in coal workers (all *p* < 0.05). A bachelor’s degree (junior college) and above, physical exercise, at least 3–4 days/week use of masks, and good ventilation and dust removal measures are protective factors for COPD among coal workers.

The random forest model is better than the CNN and logistic models in assessing COPD risk in coal workers. The COPD risk scoring system was constructed based on the random forest model that has better discriminatory ability.

## Figures and Tables

**Figure 1 ijerph-20-03655-f001:**
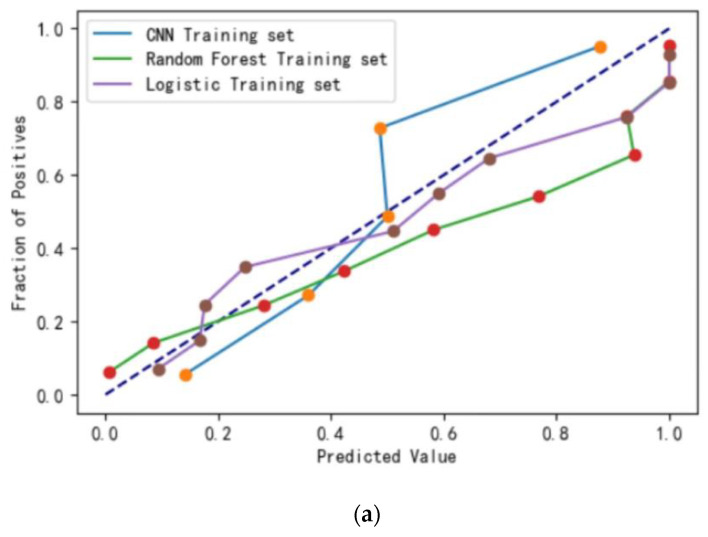
(**a**) The calibration curve training set. (**b**) The calibration curve of test set. (**c**) The calibration curve of validation set.

**Figure 2 ijerph-20-03655-f002:**
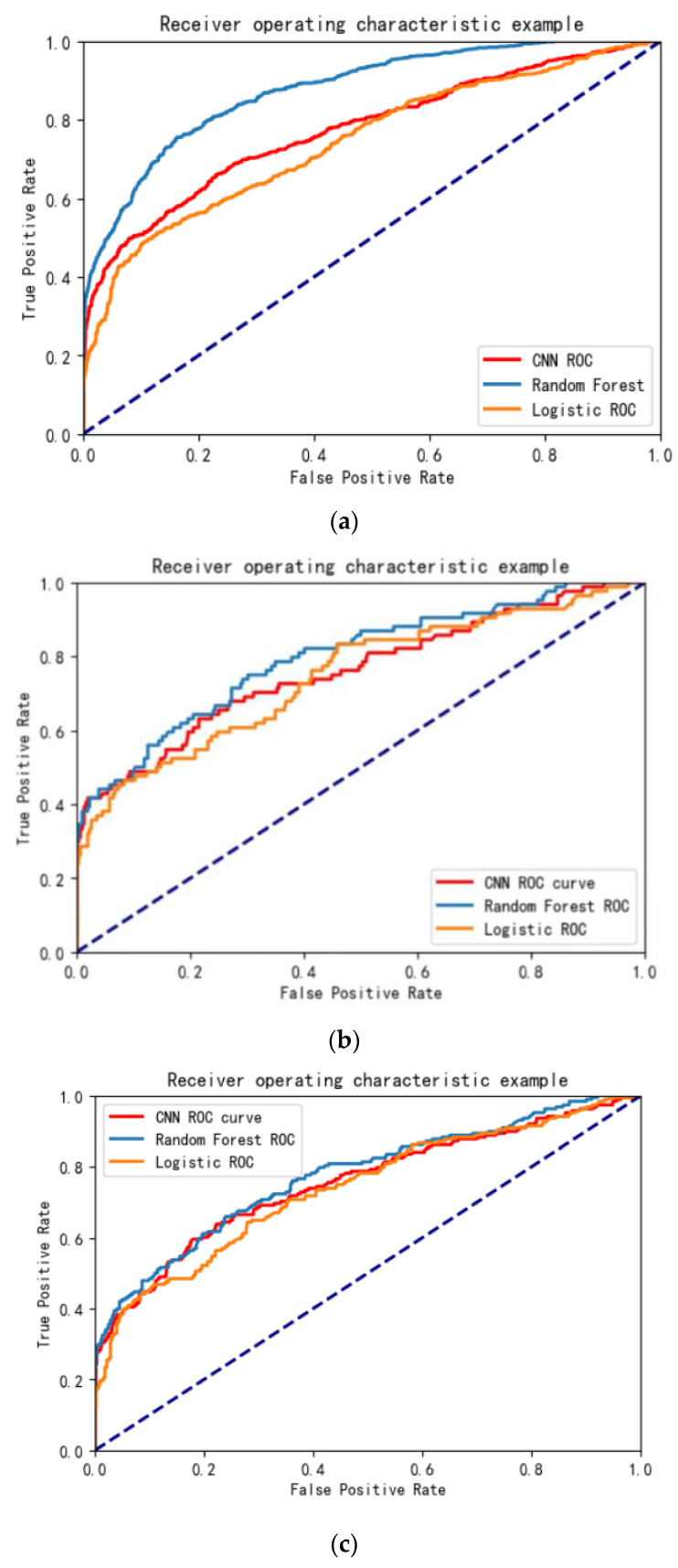
(**a**) The ROC curve of training set. (**b**) The ROC curve of test set. (**c**) The ROC curve of validation set.

**Figure 3 ijerph-20-03655-f003:**
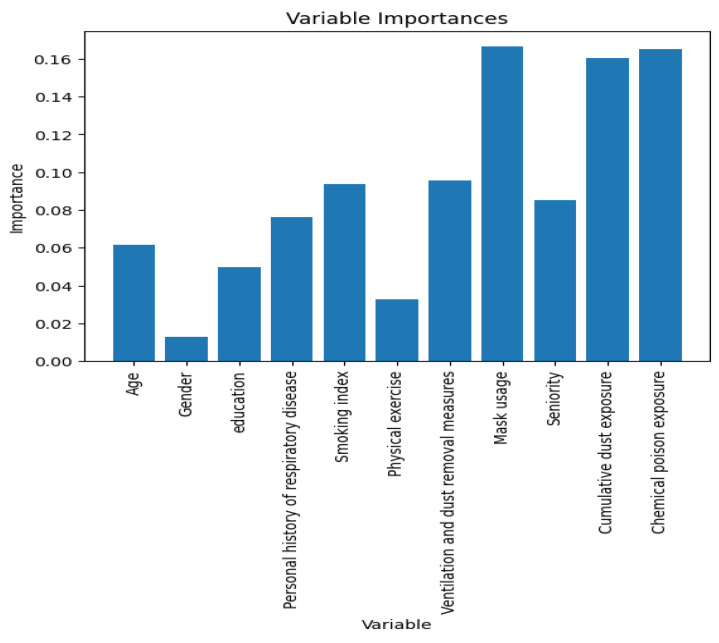
Importance ranking of predictor variables for the random forest model.

**Figure 4 ijerph-20-03655-f004:**
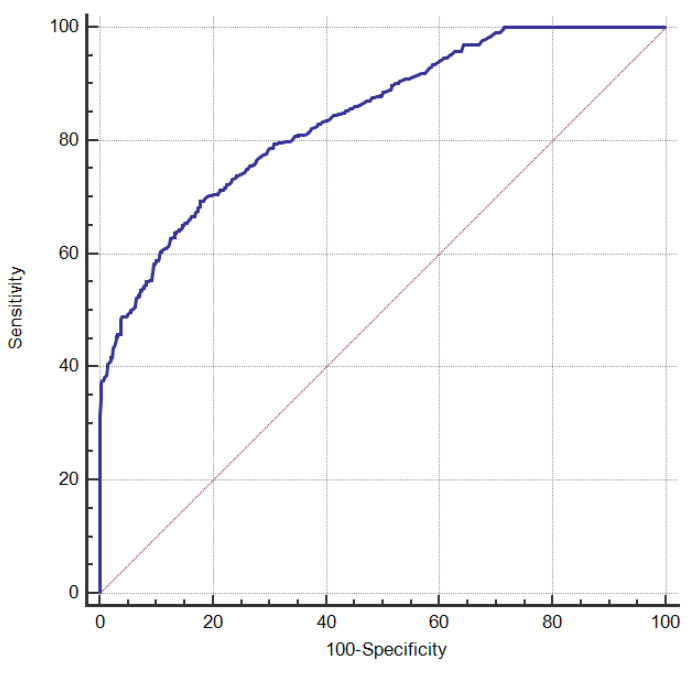
ROC curve created by COPD risk scoring system.

**Figure 5 ijerph-20-03655-f005:**
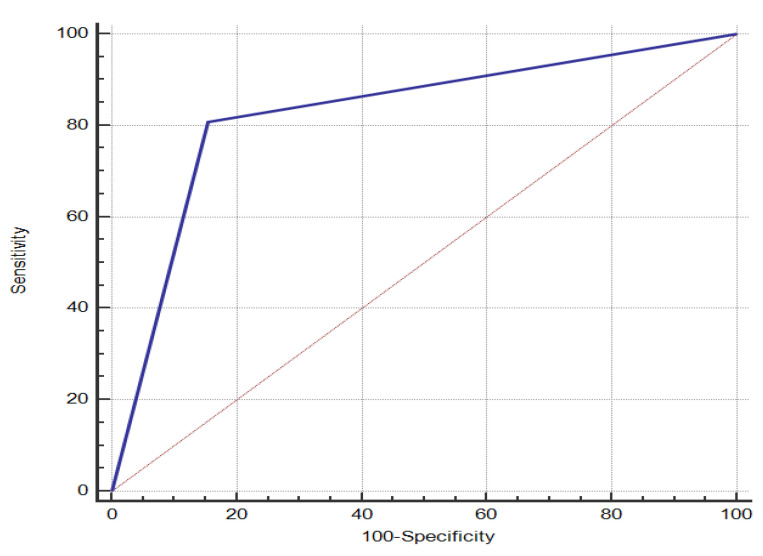
ROC curve for COPD risk scoring system validation.

**Table 1 ijerph-20-03655-t001:** Analysis of the relationship between coal workers’ general demographic characteristics and COPD.

Variable	Classify	Number	COPD		χ^2^	*p*
			Number	Proportion (%)		
Age	<30	341	30	8.8	93.746	<0.001
	30~	1931	400	20.7		
	40~	1079	280	25.9		
	50~	604	208	34.4		
Gender	Female	283	22	7.8	40.755	0.044
	Male	3672	896	24.4		
Marital status	Unmarried	149	27	18.1	5.139	0.077
	Married	3748	872	23.3		
	Others	58	19	32.8		
Education	Junior high school and below	1751	495	28.3	66.413	<0.001
	High School/technical secondary school	1222	280	22.9		
	College and above	982	143	14.6		
Household income	<5000	760	195	25.7	6.767	0.034
	5000~	2600	606	23.3		
	10,000~	595	117	19.7		
BMI (kg/m^2^)	<18.5	83	26	31.3	11.083	0.011
	18.5~	1234	307	25.7		
	24~	1723	404	23.4		
	28~	915	181	23.2		

**Table 2 ijerph-20-03655-t002:** Analysis of the relationship between coal workers’ health status and COPD.

Variable	Classify	Number	COPD		χ^2^	*p*
			Number	Proportion (%)		
Diabetes	Yes	364	97	26.6	2.657	0.103
	No	3591	821	22.9		
Hypertension	Yes	339	70	20.6	1.366	0.243
	No	3616	848	23.5		
Personal history of respiratory disease	Yes	1238	392	31.7	72.242	<0.001
	No	2717	526	19.4		

**Table 3 ijerph-20-03655-t003:** Analysis of the relationship between coal workers’ behavioral lifestyle and COPD.

Variable	Classify	Number	COPD		χ^2^	*p*
			Number	Proportion (%)		
Smoking index	0	1505	222	14.8	112.051	<0.001
	1~	890	219	24.6		
	100~	722	206	28.5		
	200~	838	271	32.3		
Drinking status	Never	1218	262	21.5	3.645	0.162
	Once	82	23	28.0		
	Now	2655	633	23.8		
Physical exercise	No	2665	710	26.6	53.949	<0.001
	Yes	1290	208	16.1		
Physical activity	Low	489	130	26.6	3.852	0.146
	Middle	316	68	21.5		
	High	3150	720	22.9		
Sleep quality	Accessibility	3296	771	23.4	2.180	0.336
	Suspicious Insomnia	469	98	20.9		
	Insomnia	190	49	25.8		
Vegetable intake	Never	103	31	30.1	12.771	0.005
	Occasionally	323	88	27.2		
	Often	843	218	25.9		
	Every day	2686	581	21.6		
Fruit intake	Never	123	35	28.5	19.334	<0.001
	Occasionally	849	234	27.6		
	Often	1069	257	24.0		
	Every day	1914	392	20.5		
Meat intake	Never	479	122	25.5	3.829	0.281
	Occasionally	2245	511	22.8		
	Often	914	202	22.1		
	Every day	317	83	26.2		
Salt	Light	736	161	21.9	0.912	0.634
	Moderate	1935	456	23.6		
	Salty	1284	301	23.4		
Soy products	Never	260	61	23.5	2.609	0.456
	Often	1539	377	24.5		
	Occasionally	1073	236	22.0		
	Every day	1083	244	22.5		

**Table 4 ijerph-20-03655-t004:** Analysis of the relationship between coal workers’ occupational harmful factors and COPD.

Variable	Classify	Number	COPD		χ^2^	*p*
			Number	Proportion (%)		
Shift situation	Never	1228	282	23.0	0.067	0.967
	Once	191	45	23.6		
	Now	2536	591	23.3		
Seniority (years)	<10	1208	171	14.2	142.547	<0.001
	10~	1748	407	23.3		
	20~	580	167	28.8		
	30~	419	173	41.3		
Cumulative dust exposure (mg/m^3^.years)	<50	546	79	14.5	232.826	<0.001
	50~	2433	439	18.0		
	100~	976	400	41.0		
Ventilation and dust removal measures	Difference	214	101	47.2	119.244	<0.001
	Ordinary	341	125	36.7		
	Good	3400	692	20.4		
Mask usage	Never	176	89	50.6	115.800	<0.001
	1–2 days/weeks	568	161	28.3		
	3–4 days/weeks	266	87	32.7		
	Every day	2945	581	19.7		
Chemical poison exposure	No	2311	323	14.0	265.997	<0.001
	Yes	1644	595	23.2		

**Table 5 ijerph-20-03655-t005:** The variable assignment method for the influencing factor.

Variable Name	Variable Meaning	Assignment Method
Y	COPD	0 = no, 1 = yes
X_1_	Age	<30 = 1, 30~ = 2, 40~ = 3, 50~ = 4
X_2_	Gender	1 = female, 2 = male
X_3_	Education	1 = junior high school and below, 2 = high school/technical secondary school, 3 = college and above;
X_4_	Household income	< 5000 = 1, 5000~ = 2, 10,000~ = 3
X_5_	Personal history of respiratory disease	0 = no, 1 = yes
X_6_	Smoking index	0 = 1, 1~ = 2, 100~ = 3, 200~ = 4
X_7_	Vegetable intake	Never = 0, occasionally = 1, often = 2, every day = 3
X_8_	Fruit intake	Never = 0, occasionally = 1, often = 2, every day = 3
X_9_	BMI	1 = <18.5, 2 = 18.5~, 3 = 24~, 4 = 28~
X_10_	Physical exercise	0 = No, 1 = yes
X_11_	Ventilation and dust removal measures	Difference = 1, ordinary = 2, good = 3
X_12_	Mask usage	0 = never, 1 = 1–2 days/weeks, 2 = 3–4 days/weeks, 3 = every day
X_13_	Seniority (years)	<10 = 1, 10~ = 1, 20~ = 2, 30~ = 3
X_14_	Cumulative dust exposure	<50~ = 1, 50~ = 1100~ = 3
X_15_	Chemical poison exposure	0 = no, 1 = yes

**Table 6 ijerph-20-03655-t006:** Multicollinearity diagnosis of independent variables.

Variable	Tolerance	VIF
Age	0.540	1.850
Gender	0.723	1.383
Education	0.874	1.144
Personal history of respiratory disease	0.965	1.037
Smoking index	0.906	1.104
Physical exercise	0.918	1.089
Ventilation and dust removal measures	0.678	1.475
Mask usage	0.710	1.409
Seniority (years)	0.554	1.804
Cumulative dust exposure	0.844	1.184
Chemical poison exposure	0.901	1.110

**Table 7 ijerph-20-03655-t007:** Results of multivariate unconditional logistic regression analysis of COPD among coal workers.

Variable	β	SE_β_	Waldχ^2^	*p*	OR (95%CI)
Age					
<30	-	-	-	-	1.00
30~	0.595	0.218	7.410	0.006	1.812 (1.181–2.781)
40~	0.645	0.216	8.888	0.003	1.833 (1.146–2.932)
50~	0.501	0.235	4.553	0.033	1.770 (1.063–2.948)
Gender					
Female	-	-	-	-	1.00
Male	1.378	0.307	20.108	*p* < 0.001	3.965 (2.172–7.247)
Education					
Junior high school and below	—	—	—	—	1.00
High school/technical secondary school	−0.186	0.099	3.529	0.060	0.830 (0.684–1.008)
College and above	−0.320	0.1123	6.756	0.009	0.726 (0.570–0.924)
Personal history of respiratory disease					
No	—	—	—	—	1.00
Yes	0.919	0.092	99.049	*p* < 0.001	2.506 (2.092–3.004)
Household income					
<5000	—	—	—	—	1.00
5000~	0.019	0.111	0.030	0.863	1.019 (0.820–1.266)
10,000~	0.055	0.156	0.125	0.724	1.057 (0.779–1.434)
Smoking index					
0	—	—	—	—	1.00
1~	0.481	0.119	16.442	*p* < 0.001	1.618 (1.282–2.041)
100~	0.542	0.124	19.265	*p* < 0.001	1.720 (1.350–2.191)
200~	0.380	0.119	10.175	0.001	1.462 (1.158–1.847)
Vegetable intake					
Never	—	—	—	—	1.00
Occasionally	0.303	0.379	0.640	0.424	1.354 (0.644–2.845)
Often	0.320	0.361	0.788	0.375	1.377 (0.679–2.793)
Every day	0.528	0.356	2.203	0.138	1.696 (0.844–3.408)
Fruit intake					
Never	—	—	—	—	1.00
Occasionally	−0.189	0.258	0.534	0.465	0.828 (0.499–1.373)
Often	−0.222	0.257	0.743	0.389	0.801 (0.484–1.326)
Every day	−0.312	0.252	1.535	0.215	0.732 (0.447–1.199)
BMI					
<18.5	—	—	—	—	1.00
18.5~	0.246	0.329	0.558	0.455	0.828 (0.499–1.373)
24~	0.141	0.326	0.186	0.666	0.801 (0.484–1.326)
28~	−0.064	0.333	0.036	0.849	0.732 (0.447–1.199)
Physical exercise					
No	—	—	—	—	1.00
Yes	−0.321	0.098	10.748	0.001	0.726 (0.599–0.879)
Ventilation and dust removal measures					
Difference	—	—	—	—	1.00
Ordinary	−1.041	0.292	12.709	*p* < 0.001	0.353 (0.199–0.626)
Good	−1.692	0.277	37.190	*p* < 0.001	0.184 (0.107–0.317)
Mask usage					
Never	—	—	—	—	1.00
1–2 days/weeks	−0.061	0.236	0.068	0.795	0.940 (0.592–1.496)
3–4 days/weeks	−0.811	0.255	10.097	0.001	0.445 (0.270–0.733)
Every day	−0.532	0.218	5.996	0.015	0.588 (0.384–0.900)
Seniority (years)					
<10	—	—	—	—	1.00
10~	0.362	0.121	8.933	0.003	1.437 (1.133–1.822)
20~	0.429	0.170	6.397	0.011	1.536 (1.101–2.143)
30~	0.597	0.195	9.408	0.002	1.817 (1.241–2.662)
Cumulative dust exposure (mg/m^3^.years)					
<50	—	—	—	—	1.00
50~	0.323	0.145	4.957	0.026	1.382 (1.039–1.837)
100~	0.801	0.157	26.102	*p* < 0.001	2.228 (1.638–3.029)
Chemical poison exposure					
No	—	—	—	—	1.00
Yes	1.091	0.092	140.818	*p* < 0.001	2.976 (2.486–3.564)

**Table 8 ijerph-20-03655-t008:** Comparison of risk assessment performance of three models.

Evaluation Indicator	Training Set			Test Set			Validation Set		
	Logistic	Random Forest	CNN	Logistic	Random forest	CNN	Logistic	Random Forest	CNN
Sensitivity (%)	78.55	86.89	77.18	66.94	81.86	75.26	62.90	82.93	74.53
Specificity (%)	85.23	92.32	87.61	79.32	87.06	83.21	81.46	84.30	82.19
Accuracy (%)	81.21	85.40	83.02	84.10	85.10	82.55	80.40	83.11	85.10
Brier score	0.14	0.10	0.14	0.15	0.13	0.15	0.13	0.11	0.13
AUC	0.76	0.88	0.78	0.74	0.82	0.76	0.75	0.78	0.77
Log Loss	0.45	0.35	0.43	0.43	0.41	0.44	0.46	0.37	0.43

**Table 9 ijerph-20-03655-t009:** Hazard score assignment scale.

Variable	Hazard Score
Age	<30 = 0, 30~ = 1.1, 40~ = 2.2, 50~ = 3.5
Gender	Female = 0, male = 1
Education	Junior high school and below = 2.5, high school/technical secondary school = 1.25, college and above = 0
Personal history of respiratory disease	No = 0, yes = 4
Smoking Index	0 = 0, 1~ = 1.5, 100~ = 3, 200~ = 4.5
Physical exercise	No = 2, yes = 0
Ventilation and dust removal measures	Difference = 4.5, ordinary = 2.25, good = 0
Mask usage	Never = 8, 1–2 days/weeks = 5.3, 3–4 days/weeks = 2.7, every day = 0
Cumulative dust exposure	<50~ = 0, 50~ = 4, 100~ = 8
Seniority	<10 = 0, 10~ = 1.3, 20~ = 2.6, 30~ = 4
Chemical poison exposure	No = 0, yes = 8

**Table 10 ijerph-20-03655-t010:** Classification results of the COPD risk scoring system.

Hazard Stratification	COPD [n(%)]		Total
	Yes	No	
<23.05	52 (19.12)	774 (84.59)	826
≥23.05	220 (80.88)	141 (15.41)	361
Total	272	915	1187

## Data Availability

Because the data are being researched and are private, the data have not been made public during the study, but can be obtained from the corresponding author upon reasonable request.

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
