# Peer review of "Development and Internal Validation of Risk Assessment Models for Chronic Obstructive Pulmonary Disease in Coal Workers"

_ijerph, 2023, doi:10.3390/ijerph20043655_

Round 1
Reviewer 1 Report
1. This paper mainly analyzes from the perspective of human. How to design the external factors (ventilation conditions, mine dust content, etc.) is not explained in the article.
2. What is the basis for establishing the influencing factors in the paper? It is not clear in the paper.
3. According to the analysis results, it is suggested to add optimization measures.
Additional comments
The paper establishes the logistic regression and convolutional neural network models.
The evaluation of occupational hazards of coal mine workers has been realized.
The idea is original and logical.However, part of the analysis is not sufficient, and the following suggestions are put forward:
1. This paper mainly analyzes from the perspective of human. How to design the external factors (ventilation conditions, mine dust content, etc.) is not explained in the article.
2. The paper data is lack of basis. What is the basis for establishing the influencing factors in the paper? It is not clear in the paper.
3. What further controls should be considered? The author should list the corresponding improvement measures according to the conclusions of this paper, and how to reduce the impact of occupational hazards of coal mine workers
Conclusion:
Reconsider after major revision (control missing in some experiments
Reviewer 2 Report
Attached.

Reviewer 3 Report
Question 1:The essay's inclusion criteria are 18~60 years old, individuals over 60 years old are excluded, and individuals over 60 years old have a higher prevalence of disease, which may be a selection bias. Can the author explain this problem?
Question 2:The authors do not give a choice of predictors for the CNN model and the random forest model, and the authors seem to use a logistic regression model to select the variables used to build the other two models.
Question3: Please confirm whether other indicators are required for the definition of COPD.
Question4: Please specify the specific classification criteria for "cumulative dust exposure" in the "Relevant Definitions".
Question5: The statistical method describes that "measurement data that follow the normal distribution are described by means and standard deviation, and non-normal distribution data are expressed by median and quartile", but the corresponding statistical analysis results are not found in the article, please confirm whether the description of the statistical method is correct.
Question6: The screening method for including model variables is not clearly expressed in "2.7 Model Establishment", can the author provide a solution to this problem?
Question7: The results of multivariate analysis in the article shows that "male" is a risk factor for disease, but this factor is not mentioned in the conclusion of the article, so please confirm by the authors.
Reviewer 4 Report
1.The authors are asked to explain whether pulmonary function measurement results can diagnose COPD.
2.Whether there is a collinearity problem between the variables included in the multivariate analysis, and whether the author has considered this problem, I have not seen an explanation of this part in the article.
3.The establishment of the model in the article does not indicate which software is used.
4.The outcome COPD is not staged in the article.
5.If the results of multi-factor analysis are inconsistent with the conclusions of the article, please confirm the results and conclusions of the article.
6.All formulas in the article are not numbered.
Reviewer 5 Report
This paper analyzes and studies the related problems of chronic obstructive pulmonary disease through neural network model, and the workload is sufficient. But there are still the following problems. 1.The overall innovation of the manuscript is not high. It is suggested that the author carefully sort out the innovation of the article by simply applying the existing data analysis methods. 2.The sample size of manuscripts is insufficient. Neural network and big data calculation need massive data support. The analysis results obtained from fewer samples do not have reference significance in the view of reviewer. 3.The manuscript is not highly referential to peers and looks more like a project report than a scientific paper.
Round 2
Reviewer 1 Report
The author has made corresponding amendments to the manuscript, which generally meets the requirements of the journal.
Reviewer 2 Report
The authors have addressed the comments from the reviewers and improved the readability of the manuscript. I recommend accepting this article in the current format.
Reviewer 5 Report
Based on the content and innovation of the manuscript, the reviewer does not believe that the manuscript has sufficient innovation and referability to make it publishable in a high quality journal.